# The CellPhe toolkit for cell phenotyping using time-lapse imaging and pattern recognition

Laura Wiggins [1,2], Alice Lord[2], Killian L. Murphy [3], Stuart E. Lacy [3], Peter J. O'Toole [1,2], William J. Brackenbury [1,2] & Julie Wilson [4] ✉

With phenotypic heterogeneity in whole cell populations widely recognised, the demand for quantitative and temporal analysis approaches to characterise single cell morphology and dynamics has increased. We present CellPhe, a pattern recognition toolkit for the unbiased characterisation of cellular phenotypes within time-lapse videos. CellPhe imports tracking information from multiple segmentation and tracking algorithms to provide automated cell phenotyping from different imaging modalities, including fluorescence. To maximise data quality for downstream analysis, our toolkit includes automated recognition and removal of erroneous cell boundaries induced by inaccurate tracking and segmentation. We provide an extensive list of features extracted from individual cell time series, with custom feature selection to identify variables that provide greatest discrimination for the analysis in question. Using ensemble classification for accurate prediction of cellular phenotype and clustering algorithms for the characterisation of heterogeneous subsets, we validate and prove adaptability using different cell types and experimental conditions.

Heterogeneity in whole cell populations is a long-standing area of interest[1–3] and previous studies have identified cell-to-cell phenotypic and genotypic diversity even within clonally derived populations[4]. The emergence of methods such as single-cell RNA sequencing has enabled characterisation of subsets within a population from gene expression profiles[5], yet these methods involve collection of data at discrete time points, missing the subtle temporal changes in gene expression on a continuous scale. Such methods exclude information on single-cell morphology and dynamics, yet cellular phenotype plays a crucial role in determining cell function[6,7], disease progression[8], and response to treatment[9]. There remains a demand for quantitative and temporal analysis approaches to describe the subtleties of single-cell heterogeneity and the complexities of cell behaviour.

Modern microscopy advancements facilitate the ability to produce information-rich images of cells and tissue, at high-throughput and of high quality. Temporal changes in cell behaviour can be observed through time-lapse imaging and features describing the cells' behaviour over time can be extracted for analysis. However, the task of identifying individual cells and following them over time is an ongoing computer vision challenge[10,11]. Initial processing requires segmentation, the detection of cells as regions of interest (ROIs) distinguished from background, and tracking, with each cell given a unique identifier that is retained over subsequent frames. Recent work using the similarity between cell metrics on consecutive frames highlighted the importance of accurate tracking to follow cell lineage[12]. Imaging artefacts vary between experiments and issues such as background noise, inhomogeneity of cell size and overlapping cells are still challenges for biomedical research[13]. Reliable cell segmentation protocols are non-deterministic and experiment-specific[14] but user-friendly software systems that use machine learning algorithms are emerging to provide objective, high-throughput cell segmentation and tracking[15,16]. Recent developments to TrackMate[17] allow the results of various segmentation

[1]York Biomedical Research Institute, University of York, York, UK. [2]Department of Biology, University of York, York, UK. [3]Wolfson Atmospheric Chemistry Laboratories, University of York, York, UK. [4]Department of Mathematics, University of York, York, UK. ✉e-mail: julie.wilson@york.ac.uk

software to be integrated with flexible tracking algorithms and provide visualisation tools to assess both segmentation and cell tracks. Although the time series for certain cell properties, such as cell area and circularity, can be displayed, the extraction and analysis of descriptive time series is not within the scope of the TrackMate software. Comparison of the tracked cells behaviour is challenging as cells are tracked for different numbers of frames with frames missing where cells leave the field of view. This has meant that analysis of any extracted features has been limited to visualisation. CellPhe interpolates the time series and then calculates a fixed number of variables that characterise each feature's time series- the features of features!

Here we present CellPhe, a pattern recognition toolkit that uses the output of segmentation and tracking software to provide an extensive list of features that characterise changes in the cells' appearance and behaviour over time. Customised feature selection allows the most discriminatory variables for a particular objective to be identified. These extracted variables quantify cell morphology, texture and dynamics and describe temporal changes and can be used to reliably characterise and classify individual cells as well as cell populations. To ensure precise quantification of cell morphology and motility, and to monitor major cellular events such as mitosis and apoptosis, it is vital that instances of erroneous segmentation and tracking are removed from data sets prior to downstream analysis methods[18]. Manual removal of such errors is heavily labour-intensive, particularly when time-lapses take place over several days. To maximise data quality for downstream analysis, CellPhe includes the recognition and removal of erroneous cell boundaries induced by inaccurate segmentation and tracking. We demonstrate the use of ensemble classification for accurate prediction of cellular phenotype and clustering algorithms for identification of heterogeneous subsets.

We exemplify CellPhe by characterising the behaviour of untreated and chemotherapy-treated breast cancer cells from ptychographic time-lapse videos. Quantitative phase images (QPI)[19–21] avoid any fluorescence-induced perturbation of the cells but segmentation accuracy can be affected by reduced differences in intensity between cells and background in comparison to fluorescent labelling. We show that our methods successfully recognise and remove a population of erroneously segmented cells, improving data set quality. Morphological and dynamical changes induced by chemotherapeutics, particularly at low drug concentration, are often more subtle than those that discriminate distinct cell types and we demonstrate the ability of CellPhe to automatically identify time series differences induced by chemotherapy treatment, with the chosen variables proving statistically significant even when not observable by eye.

The complexities of heterogeneous drug response and the problem of drug resistance further motivate our chosen application. The ability to identify discriminatory features between treated and untreated cells can allow automated detection of "non-conforming" cells such as those that possess cellular drug resistance. Further investigation of such features could elucidate the underlying biological mechanisms responsible for chemotherapy resistance and cancer recurrence. We validate the adaptability of CellPhe with both a different cell type and a different drug treatment and show that variables are selected according to experimental conditions, tailored to properties of the cell type and drug mechanism of action.

CellPhe is available on GitHub as an R package with a user-friendly interactive GUI that allows completely unbiased cell phenotyping using time-lapse data from fluorescence imaging as well as ptychography. A working example guides the user through the complete workflow and a video demonstrating the GUI is also provided.

## Results
### Overview of CellPhe
CellPhe is a toolkit for the characterisation and classification of cellular phenotypes from time-lapse videos, a diagrammatic summary of

CellPhe is provided in Fig. 1. Experimental design is determined by the user prior to image acquisition where seeded cell types and pharmacology are specific to the user's own analysis. Example uses are discrimination of cell types (e.g, neurons vs. astrocytes), characterisation of disease (e.g., healthy vs. cancer), or assessment of drug response (e.g., untreated vs. treated). The user can then time-lapse image cells for the desired amount of time, using an imaging modality of their choice. Once images are acquired and segmentation and tracking of cells are complete, cell boundary coordinates are exported and used for calculation of an extensive list of morphology and texture features. These together with dynamical features and extracted time series variables are used to aid removal of erroneous segmentation by recognition of error-induced interruption to cell time series. Once all predicted segmentation errors have been removed from data sets, feature selection is performed and only features providing separation above an optimised threshold are retained. This identifies a list of most discriminatory features and allows the user to explore biological interpretation of these findings. The extracted data matrices are then used as input for ensemble classification, where the phenotype of new cells can be accurately predicted. Furthermore, clustering algorithms can be used to identify heterogeneous subsets of cells within the user's data, both inter- and intra-class.

The remaining results exemplify the use of CellPhe with a biological application, characterisation and classification of chemotherapeutic drug response. We look at each of the CellPhe stages in detail (segmentation error removal, feature selection, ensemble classification and cluster analysis) and demonstrate that each step provides interpretable, biologically relevant results to answer experiment-specific questions and aid further research.

## CellPhe application: characterising chemotherapeutic drug response
The 231Docetaxel data set, obtained from multiple experiments involving MDA-MB-231 cells, both untreated and treated with 30 μM docetaxel, is the main data set used to demonstrate our method. We show that the same analysis pipeline can be applied to other data sets by considering both a different cell line, MCF-7, in the MCF7Docetaxel data set, and a different drug, doxorubicin, with the 231Doxorubicin data set. In each case, we remove segmentation errors, as described in Section 2.5, before using feature selection (Section 2.6) to identify discriminatory variables tailored to the particular data set. We show that different variables are chosen depending on the inherent nature of the cell line and the effect of the drug in question. Using these features in classification algorithms, we characterise and compare the behaviour over time of untreated and treated cells.

## Segmentation error removal
We improve the quality of our data sets prior to untreated vs. treated cell classification by automating detection of segmentation errors and optimising the exclusion criteria of predicted errors.

Comparison of time series for cells with and without segmentation errors showed many of our features to be sensitive to such errors, motivating the need to remove these cells prior to treatment classification. Size metrics, such as volume, were particularly affected by segmentation errors as under- or over-segmentation could result in halving or doubling of cell volume respectively (Fig. 2a, b). Such noticeable disruption to the time series of several features suggested that reliable detection of segmentation errors would be possible.

After excluding 62 instances identified as tracked cell debris, a training data set for MDA-MB-231 cells (from the 231Docetaxel data set), was obtained, consisting of 1701 correctly segmented cells and 241 cells with segmentation errors. The number of cells in the segmentation error class was doubled using SMOTE and the resulting data set with 2184 observations used for the classification of segmentation errors as described in Section 2.5. The MDA-MB-231 cells (from

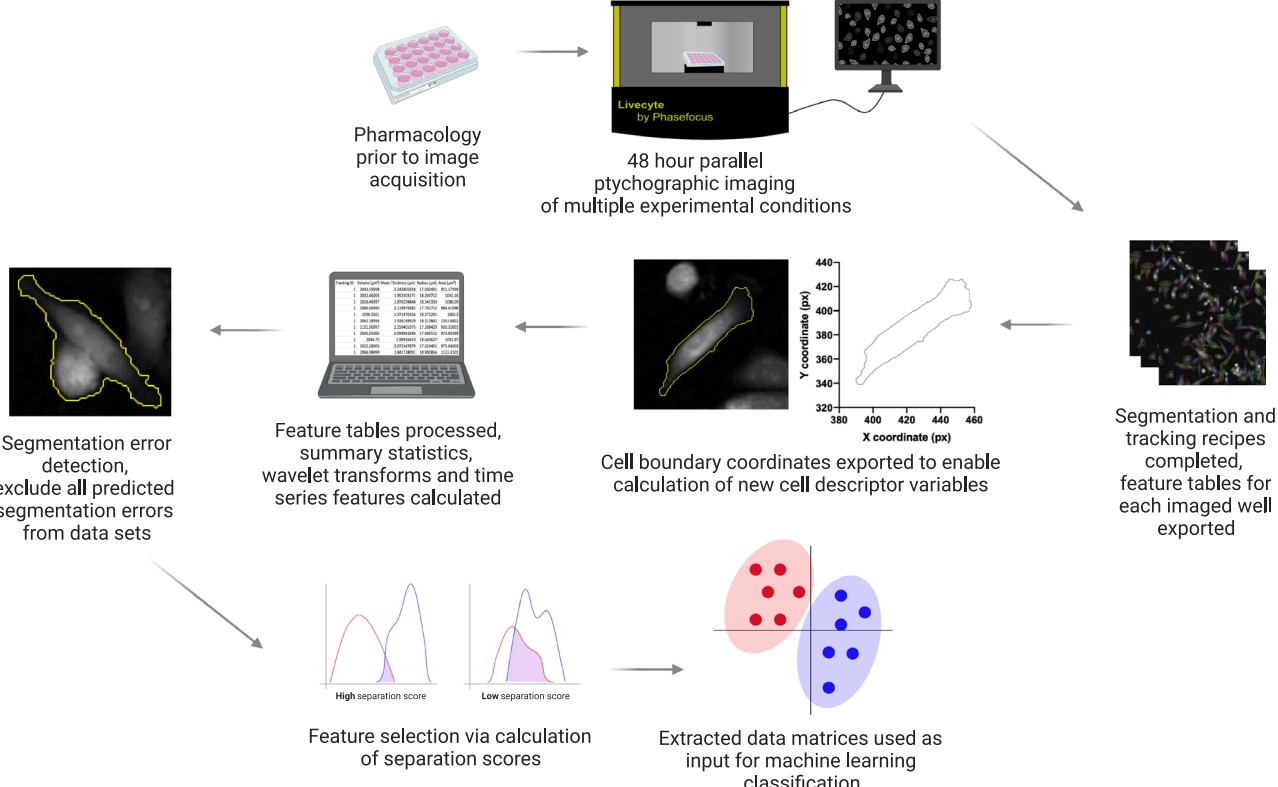

**Fig. 1 | Summary of the CellPhe toolkit.** Following time-lapse imaging, acquired images are processed and segmentation and tracking recipes implemented. Cell boundary coordinates are exported, features extracted for each tracked cell and the time series summarised by characteristic variables. Predicted segmentation errors are excluded and optimised feature selection performed using a threshold on the class separation achieved. Finally, multiple machine learning algorithms are combined for classification of cell phenotype and clustering algorithms utilised for identification of heterogeneous cell subsets. This figure was created with BioRender.com.

231Docetaxel and 231Doxorubicin, both untreated and treated) that were not used for training formed independent test sets (Table 1).

A total of 223 of the 1478 cells in the 231Docetaxel test set were predicted to be segmentation errors. Of these, 217 were confirmed by eye to be true segmentation errors, most of which were due to under- or over-segmentation throughout their time series. Other segmentation issues observed included background pickup, cells swapping cell ID, and cells repeatedly entering and exiting the field of view, all of which result in problem time series (Fig. 2c, d). Of the remaining six cells that were misclassified as segmentation errors, one was a large cell and the other five were cells tracked before, during, and after attempted mitosis. Further investigation showed that the removal of these cells did not exclude an important subset from the data.

This classifier was also used to identify a further 78 segmentation errors from the 955 cells in the 231Doxorubicin data set, all 78 were confirmed by eye to be true segmentation errors (Table 1). It was necessary to train a new classifier for MCF-7 segmentation error detection due to differences between the cell lines. In this case, 308 correctly segmented cells and 192 segmentation errors were identified by eye. After applying SMOTE to double the number of segmentation error observations, a classifier was trained with the resulting 692 observations as described in section 2.5. 188 cells in the MCF7Docetaxel data set (848 cells in total) were classified as segmentation errors. 185 of these cells were confirmed by eye to be true segmentation errors, the remaining three were large cells or cells tracked before, during and after attempted mitosis.

As decision trees are used in the identification of segmentation errors, our feature selection is not required. However, we still calculated separation scores for the MDA-MB-231 training data to investigate the effect of such errors. As might be expected, volume was most affected, with segmentation errors resulting in larger standard deviation, ascent and maximum value. Other features with high separation scores included area as well as spatial distribution descriptors with the highest thresholds, features that detect the clustering of high intensity pixels, characteristic of cell overlap and over-segmentation (Fig. 2e). Analysis of the trained decision trees showed that a combination of size, shape, texture and density variables frequently formed the most important features for detecting segmentation errors with MDA-MB-231 cells, see Fig. 2f for an example.

For the MCF7Docetaxel data set, velocity was found to be important in determining whether or not a cell experienced segmentation errors in addition to texture and shape variables. The cell centroid, used to determine position and hence velocity, is affected by boundary errors and so high velocity, uncharacteristic of MCF-7 cells, is a good indication of segmentation error for these cells.

## Feature selection

For the 231Docetaxel data set, the calculation of separation scores identified variables that provided good discrimination between untreated MDA-MB-231 cells and those treated with 30 μM docetaxel. As separation scores do not provide information on how these variables work in combination, we performed Principal Component Analysis (PCA) to explore relationships between discriminatory variables.

Differences in the appearance of MDA-MB-231 cells induced by docetaxel treatment were observed by eye from cell time-lapses. Untreated cells displayed a spindle-shaped morphology (a circular cross-section with tapering at both ends), with contractions and protrusions facilitating migration. Cells that received treatment were generally dense and spherical, and increased in size following a failed attempt at cytokinesis (Fig. 3a, b). Discriminatory features identified by

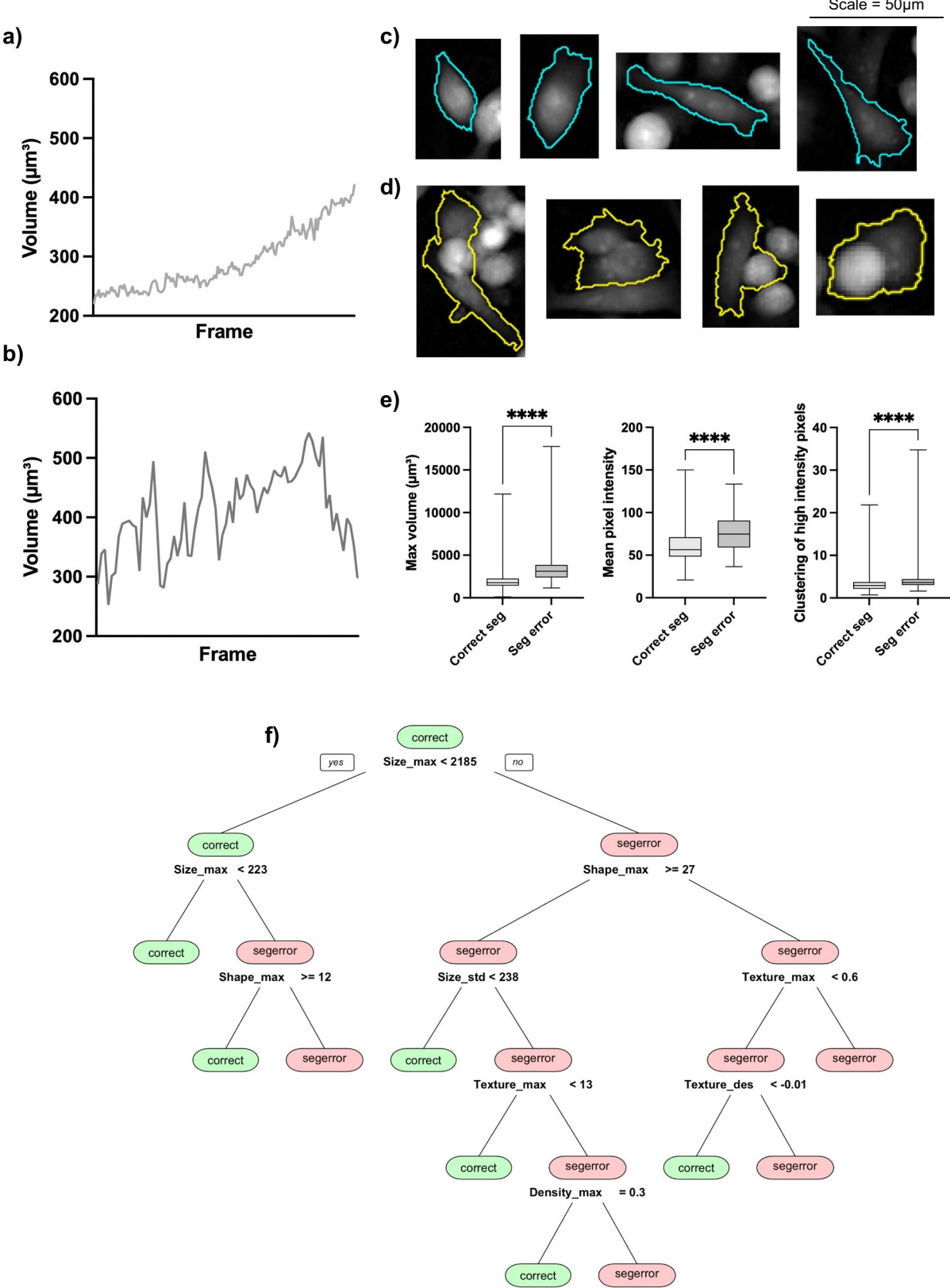

**Fig. 2 | Characterisation of segmentation errors. a** Volume time series for a correctly segmented cell and **b** a cell experiencing segmentation errors, demonstrating greater fluctuation in volume when a cell experiences segmentation errors. Examples of test set cells classified as **c** correct segmentation and **d** segmentation error. Note that the scale bar applies to all cell images in **c**, **d**. **e** Box and whisker plots of features that are significant for identifying segmentation errors in the 231Docetaxel training set (****: $p < 0.0001$). The median value is shown by the line within the box representing the interquartile range (IQR), with lines at the 25th and 75th percentile, whiskers extend to the maximum and minimum values. $p$ values were calculated using a two-tailed, non-parametric Mann-Whitney $U$ test at the 95% confidence interval. $n = 1702$ and 241 for correctly segmented cells and segmentation errors respectively. **f** A representative 231Docetaxel trained decision tree, demonstrating how size, shape, texture and density are used in combination to make classifications. Source data for **e**, **f** are provided in the Source Data file.

calculation of separation scores were consistent with differences observed by eye, the 100 variables that achieved greatest separation are shown in Fig. 3c. Texture, shape, and size variables provided greatest discrimination of untreated and treated cells. Untreated cells experienced increased elongation throughout the time-lapse and displayed irregular, spindle-shaped morphology in comparison to the generally spherical appearance of treated cells. Furthermore, separation scores highlighted differences in the texture of cells, with intensity quantile metrics characterising changes in granularity of cells induced by drug treatment.

### Table 1 | Segmentation error prediction on the test data

| Data set | TP | FP |
|---|---|---|
| 231Docetaxel (1478) | 217 | 6 |
| 231Doxorubicin (955) | 78 | 0 |
| MCF7Docetaxel (848) | 185 | 3 |

The number of correctly classified segmentation errors (True Positives, TP) and the number of correctly segmented time series incorrectly classified as segmentation errors (False Positives, FP) are shown. The number of cells in each test data before segmentation error removal is shown in parentheses.

Principal Component Analysis (PCA) demonstrated that the main variance within the data arises due to class differences, with separation of classes observed across PC1 which explains 66% of the total variance (Fig. 3d). The dispersion of points within the scores plot illustrates heterogeneity of cells both inter- and intra-class. The non-conformity of some cells, for example, treated cells behaving as untreated cells, is demonstrated by points clustering within the opposite class. Analysis of PCA loadings highlighted increased ascent, descent, and standard deviation for untreated cells, as can be observed from the PCA biplot in Fig. 3e. Although descent variables appear to have opposite loadings to all other variables, in fact, this is only due to their negative values. As the majority of untreated cells had negative PC1 scores we deduced that greater standard deviation, ascent and descent of features for untreated cells indicates that these cells experience increased fluctuation throughout their time series. As treated cells mainly had positive PC1 scores, they experience less fluctuation throughout their time series and instead display greater stability. Identified differences in feature time series are visualised in Fig. 3f.

We assessed the adaptability of our feature selection method by calculating separation scores for both a different cell line and a different treatment, using PCA to evaluate the main sources of variance.

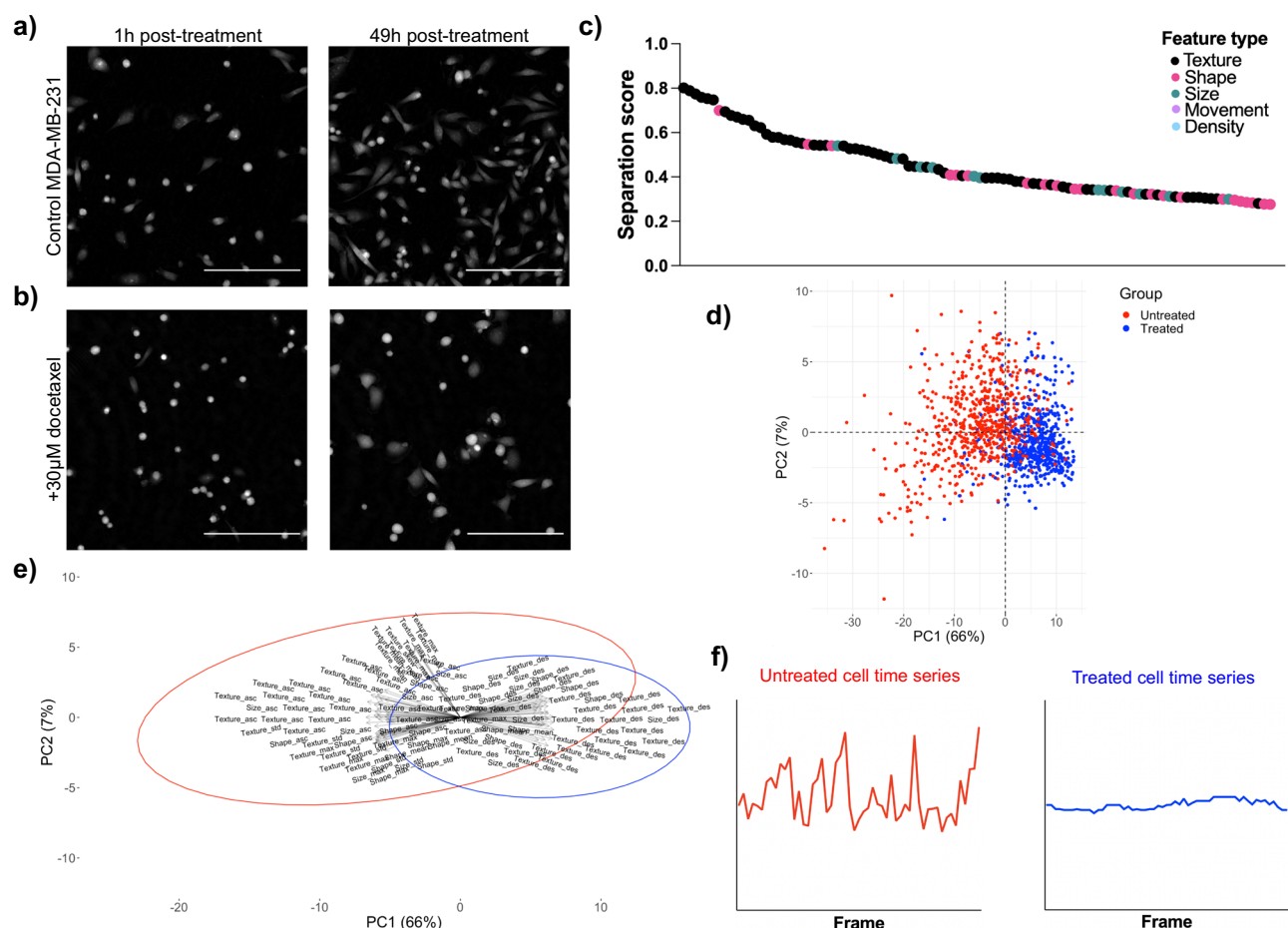

**Fig. 3 | Discrimination between treated and untreated cells for MDA-MB-231 with docetaxel.** Images taken from cell time-lapses of **a** untreated MDA-MB-231 cells and **b** 30 μM docetaxel treated MDA-MB-231 cells. Scale bar = 200 μm. Increased cell count at 49h post-treatment demonstrates healthy proliferation of untreated cells. Static cell count at 49h for treated cells is a result of cell cycle arrest and failed cytokinesis, leading to enlarged cell phenotype. **c** Features with the top 100 highest separation scores, colour-coded according to feature type. Texture, shape, and size features provide greatest separation. **d** Principal Component Analysis (PCA) scores plot with points colour-coded according to true class label. Observable separation of classes along PC1 demonstrates that the greatest source of variance within the data arises due to class differences. Only features with the 100 highest separation scores were included in PCA. **e** PCA biplot demonstrating how features with the 100 highest separation scores work in combination to discriminate between untreated and 30 M docetaxel-treated MDA-MB-231 cells. Greater ascent and descent can be observed for untreated cells, indicating greater activity across a range of features for untreated cells. **f** Representative feature time series plots for untreated and 30 μM docetaxel-treated MDA-MB-231 cells. Untreated cells experience greater fluctuation within their time series in comparison to treated cells where activity is more stabilised. Source data for **c–e** are provided in the Source Data file.

We compared MCF-7 cells treated with 1 μM docetaxel with untreated MCF-7 cells, and MDA-MB-231 cells that were treated with 1μM doxorubicin with untreated MDA-MB-231 cells and found that changes in the morphology and motility of cells upon treatment were both drug and cell-line specific with different variables selected (Fig. 4).

As was observed within the 231Docetaxel time-lapses, cells increased in size due to failed cytokinesis. However, MCF-7 cells maintained a polygonal, epithelial-like morphology following treatment similar to that of the untreated population. Conversely, remarkable differences in cellular dynamics were observed within the 231Doxorubicin data set, with motility of cells being severely hindered following treatment, particularly after the 24-hour time point. Only subtle differences in size and morphology of cells were observed by eye, with doxorubicin treated cells appearing slightly enlarged as a result of cell cycle arrest. Both untreated and treated sets contained examples of cells in G1 and G2, hence varied cell morphology can be observed within both (elongated and adherent cells in G1, round and dense morphology of cells in G2.)

The 100 variables that achieved greatest separation for each of the MCF7Docetaxel and 231Doxorubicin data sets are shown in Fig. 4c, d. Density variables were highly discriminatory for untreated and docetaxel treated MCF-7 cells, characterising decreased proliferation and cell-cell adhesion induced by drug treatment. Size, shape and texture variables were also identified as most discriminatory with variables such as length, width and area characterising the enlarged cell shape of treated cells. Spatial distribution variables were chosen for several intensity thresholds, demonstrating differences in the clustering of pixels, following docetaxel treatment. As was observed by eye, movement features formed the majority of discriminatory variables for the 231Doxorubicin data set, with untreated cells having greater velocity, tracklength and displacement than treated cells. Differences in movement were also described through density ascent and descent, as cell density fluctuated more for untreated cells due to the increased likelihood of passing neighbouring cells when migrating. Subtle differences in cell shape and size observed by eye upon doxorubicin treatment were described by changes in rectangularity, width and radius variables. Notably both data sets received lower separation scores than the 231Docetaxel data set, with 231Doxorubicin having the lowest. This effectively provides a measure of class similarity, with high separation scores for 231Docetaxel indicative of significant changes to cells upon treatment and low separation scores for 231Doxorubicin suggesting these changes are more subtle.

PCA scores plots obtained with the selected features are shown in Fig. 4d. Differences between classes can be observed for the MCF7Docetaxel data set, with separation of classes along PC1 (40% of the total variance) and PC2 (13% of the total variance). The PCA scores plot for 231Doxorubicin shows the greatest source of variance to be due to class differences, with separation of classes along PC1 (49% of the total variance). All PCA scores plots demonstrated the potential to characterise untreated and treated cell behaviour, with feature-selected variables providing good distinction of classes which was improved by using variables in combination.

## Classification of treated and untreated cells

We found that the distribution of separation scores differed for each data set, with the 231Docetaxel set having the greatest number of variables achieving high separation, followed by MCF7Docetaxel and 231Doxorubicin generally having much lower separation scores (Fig. 5a & b). Optimal separation thresholds of 0.075, 0.025 and 0.025 were obtained for 231Docetaxel, MCF7Docetaxel and 231Doxorubicin respectively, resulting in 437, 539 and 442 variables (of a possible 1111) being selected for classifier training.

Having chosen an optimal separation threshold, we trained an ensemble classifier for each data set as described in Section 2.6. Classification accuracy scores for training and test sets obtained using

our ensemble classifier are provided in Table 2. Through visual inspection, we found that misclassifications formed subsets of cells whose behaviour deviated from the behaviour of the main population, we call this subset non-conforming. (Fig. 5c). For untreated cells, we found that healthy, proliferating cells were correctly classified whereas less motile cells, cell debris or large, non-motile mutant cells were instead classified as treated. For treated cells, we found that cells experiencing the drug-induced phenotypic differences identified through feature selection were classified as treated. However, treated cells displaying behaviour similar to that of an untreated cell, such as increased migration or fluctuation and elongation in cell shape, and were classified as untreated (Fig. 5d).

We found that the proportion of non-conforming treated cells, those classified as untreated, decreased as drug concentration increased for all three data sets (Fig. 5e). To explore the connection between the proportion of non-conforming treated cells and the population drug response of each treated set, we considered the total volume growth rate at each drug concentration in relation to the percentage of cells predicted as untreated (Fig. 5f). We found that the overall growth rate decreased with increased drug concentration due to more cells responding at higher concentrations. This correlated positively with the percentage of cells predicted as untreated, with a greater percentage of cells predicted as untreated for high volume growth rate with proliferation still occurring.

## Subset identification

Classification accuracy scores for the untreated and treated cell populations were imbalanced across all three of the data sets (Table 2). Imbalance of classification accuracy scores in binary classification is often a result of hidden stratification[22], where poor performance of one class is a result of misclassifications of important, unlabelled subsets. To investigate this phenomenon we performed hierarchical clustering on 231Docetaxel treated cells and the obtained dendogram is provided in Fig. 6a, b, with examples of cells from each cluster.

Figure 6 c shows the distribution of mean volumes for each cluster in comparison to the untreated MDA-MB-231 population. Clusters 1 and 2 span a similar range of volumes to the untreated set, whereas clusters 3 and 5 have greater mean volumes. Cluster 4 is formed primarily of cell debris as a result of cell death with mean volumes much lower than those of the untreated set.

Cells in the same cluster share similar properties and morphological differences between clusters of different cell cycle states can be observed. For example cells in clusters 1 and 2 are much smaller and brighter than cells in clusters 3 and 5 as the cells are heading towards attempted mitosis, confirmed by visual inspection of cell time-lapses, and hence resemble untreated mitotic cells. The PCA biplot in Fig. 6d shows how variables work in combination to determine cell clusters. Clusters 1 and 2 are generally bright and spherical, similar to a mitotic-treated cell, as these cells are tracked prior to failed cytokinesis. Cells that have attempted to split, clusters 3 and 5, are larger, longer, wider and display greater irregularity in shape. These cells become less dense and are often multinucleated resulting in changes to texture features. Cell debris is best distinguished by granularity, hence texture metrics are fundamental in identifying these instances.

Clusters also spanned a range of mean cell volumes beyond those of the untreated set when hierarchical clustering was repeated for MCF7Docetaxel-treated cells. However, this was not the case for 231Doxorubicin-treated cells and therefore k-means clustering was used to explore the connection between misclassifications and hidden subsets in the 231Doxorubicin treated cell test set. Two distinct clusters were obtained (Fig. 6e), cluster 1 was formed of 33 cells and cluster 2 of 32 cells. We calculated classification accuracy scores for the two clusters individually and found that 91% of cells in cluster 1 were correctly classified as treated but only 31% in cluster 2 (Fig. 6f). The increased migration and fluctuation in shape of cells in cluster 2 mean

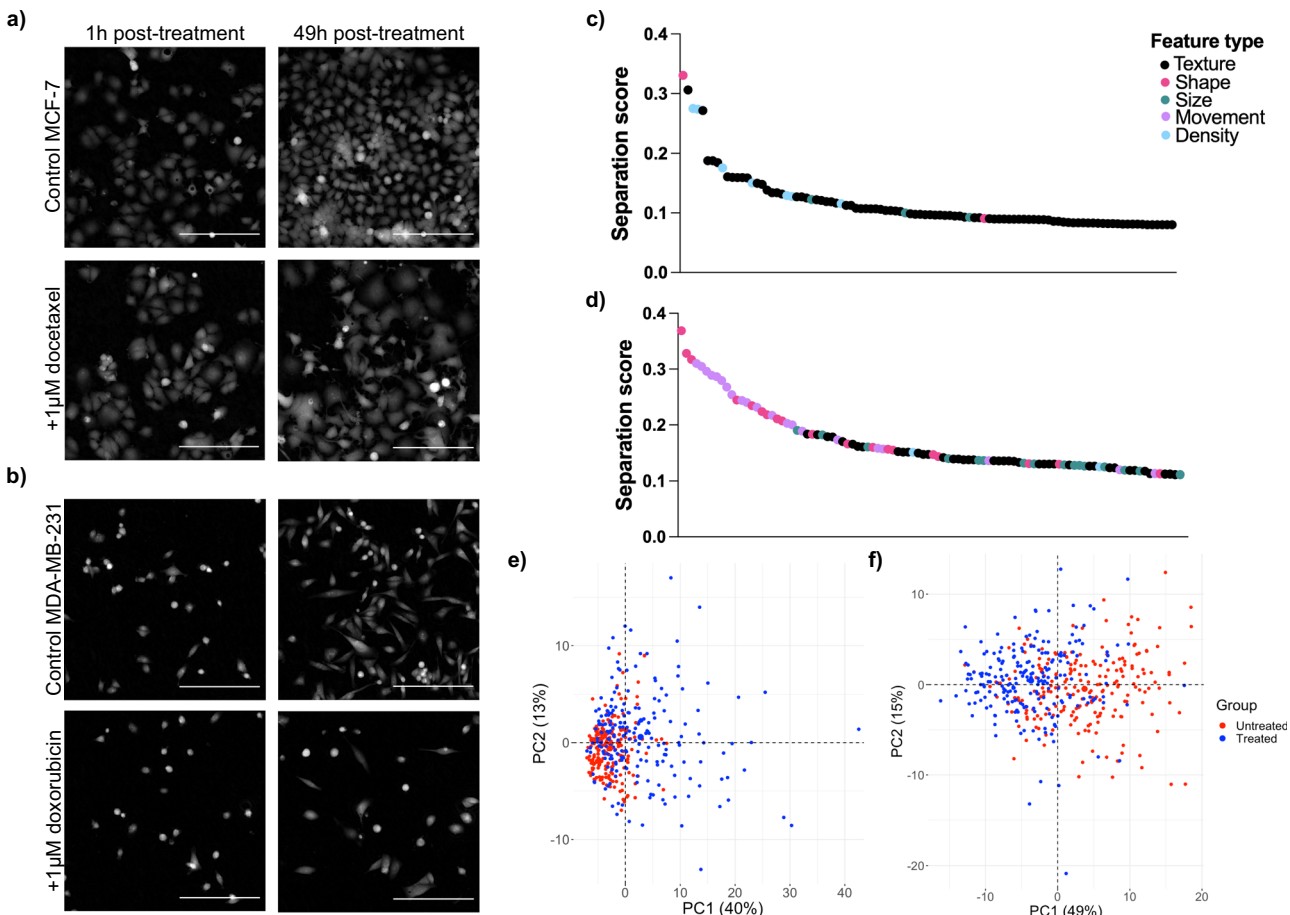

**Fig. 4 | Discrimination between treated and untreated cells for MCF-7 with docetaxel and MDA-MB-231 with doxorubicin.** Images taken from cell time-lapses of **a** untreated and 1 μM docetaxel treated MCF-7 cells and **b** untreated and 1 μM doxorubicin treated MDA-MB-231 cells. Scale bar = 200 μm. Differences in cell count following treatment can be observed for both due to cell cycle arrest induced by docetaxel or doxorubicin respectively. Docetaxel treated MCF-7 cells display enlarged cell phenotype at the 49h time point due to failed cytokinesis. In comparison, differences in morphology are more subtle for doxorubicin treated MDA-MB-231 cells at the 49h time point. Features with the top 100 highest separation scores, colour-coded according to feature type for **c** MCF7Docetaxel, where cell density and texture provide greatest separation, and **d** 231Doxorubicin where shape and movement features provide greatest separation. Principal Component Analysis (PCA) scores plot with points colour-coded according to true class label for **e** MCF7Docetaxel and for **f** 231Doxorubicin. Only features with the 100 highest separation scores were included in PCA. Source data for **c**–**e**, **f** are provided in the Source Data file.

these cells have greater similarity to the untreated population (Fig. 6g). These non-conforming treated cells form the majority of treated cell misclassifications in the 231Doxorubicin test set and highlight the presence of heterogeneous subsets within a population.

Notably there was a greater number of misclassifications for untreated MCF-7 cells in comparison to the docetaxel-treated set. Cluster analysis demonstrated the presence of heterogeneous subsets within the untreated population, with one cluster, in particular, consisting mainly of misclassified cells (Supplementary Figure 1). Texture metrics discerned this cluster from other untreated cell clusters, containing several instances of cell debris that were understandably classified as non-conforming. Other cells within this cluster shared similarities in texture to cell debris.

### Compatibility with fluorescence images and TrackMate

TrackMate-Cellpose[17] was used to demonstrate the compatibility of CellPhe with outputs obtained from alternative segmentation and tracking software and show that CellPhe extends to fluorescence time-lapse imaging. Ptychographic and fluorescence time-lapse images of untreated and docetaxel-treated MDA-MB-231 cells stably expressing dsRed were acquired in parallel (Fig. 7a). Cell segmentation from the fluorescence images was performed using Cellpose and segmented cells were then tracked using TrackMate resulting in 123 cell tracks of greater than or equal to 50 frames (Fig. 7b). The resulting folders of cell ROIs and TrackMate feature tables were used as input for CellPhe to extract single-cell phenotypic metrics to describe cell behaviour over time. An optimal separation threshold of 0.3 was determined for discrimination between untreated and treated cells, with 231 variables achieving separation scores greater than the threshold (Fig. 7c). As observed with the phase images, size, shape, and texture variables provide the greatest separation, with cell density amongst the most discriminatory variables. Good separation of untreated and treated cells can be observed within the PCA scores plot in Fig. 7d, supporting the use of CellPhe for cell phenotyping from fluorescence images.

## Discussion

The CellPhe toolkit complements existing software for automated cell segmentation and tracking, using their output as a starting point for bespoke time series feature extraction and selection, cell classification and cluster analysis. Erroneous cell segmentation and tracking can significantly reduce data quality but such errors often go undetected and can negatively influence the results of automated pattern recognition. CellPhe's extensive feature extraction followed by customised feature selection not only allows the characterisation and classification of cellular phenotypes from time-lapse videos but provides a method for the identification and removal of erroneous cell tracks prior to

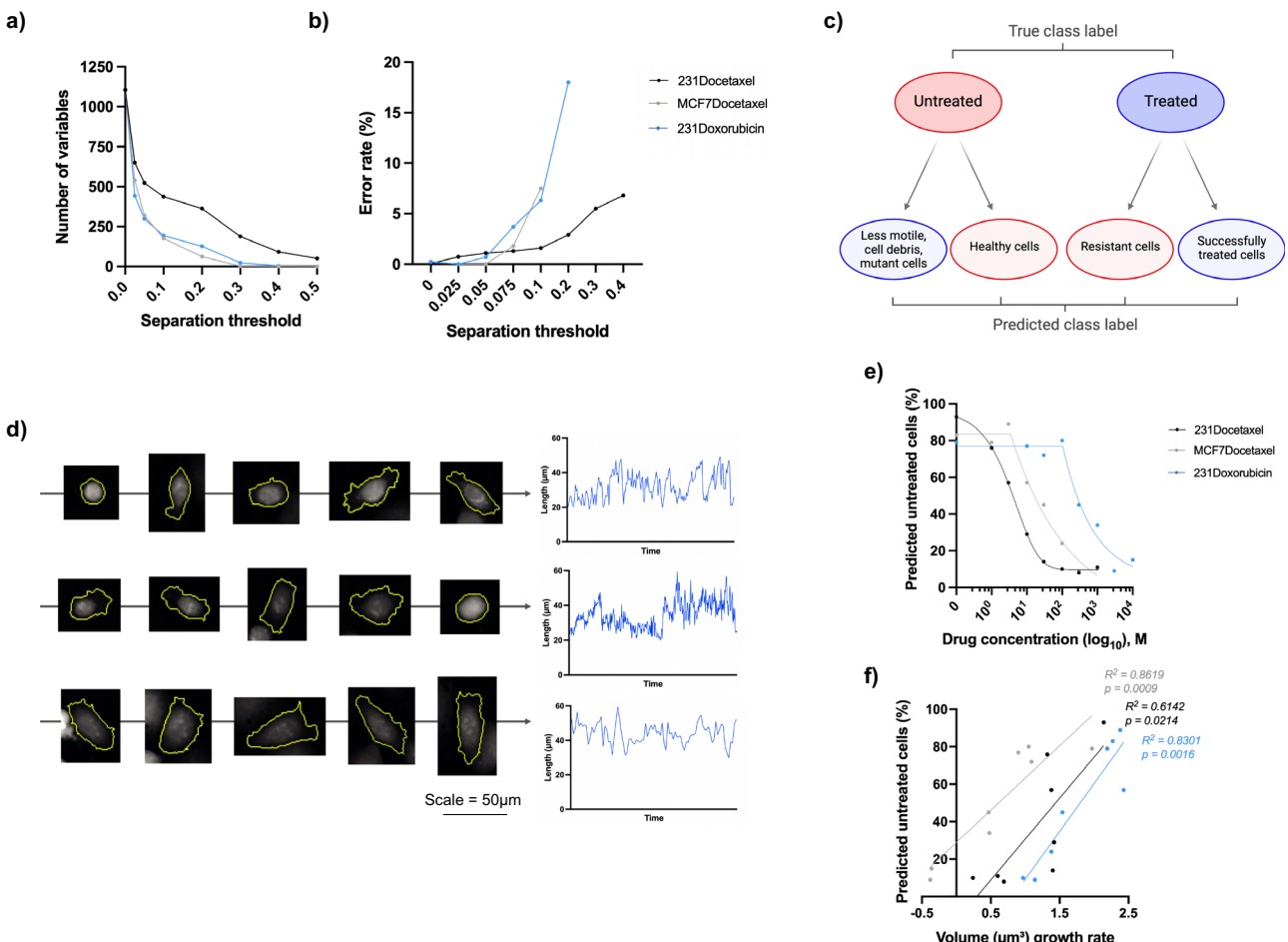

**Fig. 5 | Analysis of misclassified cells. a** The number of variables with separation scores above different thresholds. A greater number of variables achieve high separation for 231Docetaxel in comparison to 231Doxorubicin and MCF7Docetaxel. **b** Optimisation of separation threshold for each data set. Thresholds of 0.075, 0.025, and 0.025 were selected for 231Docetaxel, MCF7Docetaxel and 231Doxorubicin respectively resulting in 437, 539 and 442 variables being used for classifier training. **c** Sub-populations within each class, colour-coded according to the ideal final classification of each sub-population. Non-conforming cells for each class form a subset of misclassified cells. **d** Examples of docetaxel treated MDA-MB-231 cells misclassified as untreated. Time-lapse images demonstrate how these cells exhibit an elongated morphology characteristic of migratory untreated cells, note that the scale bar applies to all cell images. Time series plots for cell length demonstrate the fluctuation in shape of these cells, typical of untreated cells. **e** The percentage of cells predicted as untreated for a range of drug concentrations ($log_{10}$ scale). For all three data sets, this percentage decreases as drug concentration increases due to a greater number of cells responding to treatment at higher concentrations. Lines were fitted using asymmetric, five parameter, non-linear regression. **f** Positive correlation between the total volume rate of growth and the percentage of cells predicted as untreated, with higher volume growth rates associated with a higher number of cells being predicted as untreated. Linear regression slopes were found to be significant ($p$ values shown). $R^2$ correlation coefficients are also provided, demonstrating positive correlation for each data set. $p$ values were calculated using an $F$-test with 6 degrees of freedom. Source data for **a**, **b**, **e**, **f** are provided in the Source Data file.

these analyses. Attribute analysis showed that different features were chosen to identify segmentation errors for different cell lines. For example, sudden increases in movement resulting from large boundary changes can indicate segmentation errors for MCF-7 cells,

## Table 2 | Ensemble classification accuracy scores for each data set

|       |                  | 231Docetaxel     | MCF7Docetaxel    | 231Doxorubicin   |
|-------|------------------|------------------|------------------|------------------|
| Train |                  | Untreated: 98%   | Untreated: 100%  | Untreated: 100%  |
|       |                  | Treated: 100%    | Treated: 99%     | Treated: 100%    |
|       |                  | Overall: 99%     | Overall: 100%    | Overall: 100%    |
| Test  |                  | Untreated: 97%   | Untreated: 83%   | Untreated: 86%   |
|       |                  | Treated: 85%     | Treated: 90%     | Treated: 66%     |
|       |                  | Overall: 94%     | Overall: 85%     | Overall: 81%     |

All percentages have been rounded to the nearest whole number. Source data are provided in the Source Data file.

contrasting with their innate low motility. On the other hand, size and texture variables provide better characterisation of the unexpected fluctuations in cell size and clusters of high-intensity pixels induced by segmentation errors for MDA-MB-231 cells. Current approaches for removal of segmentation errors are subjective and labour-intensive, requiring manual input of parameters such as expected cell size that need to be fine-tuned for different data sets. CellPhe provides an objective, automated approach to segmentation error removal with the ability to adapt to new data sets.

For cell characterisation, we have shown that CellPhe's feature selection method is able to adapt to different experimental conditions, providing discrimination between untreated and treated groups of two different breast cancer cell lines (MDA-MB-231 and MCF-7) and two different chemotherapy treatments (docetaxel and doxorubicin). The discriminatory variables identified here coincide with previously reported effects of docetaxel or doxorubicin treatment and can be interpreted in terms of the mechanism of action of each drug. Previous studies have identified a subset of polyploid, multinucleated

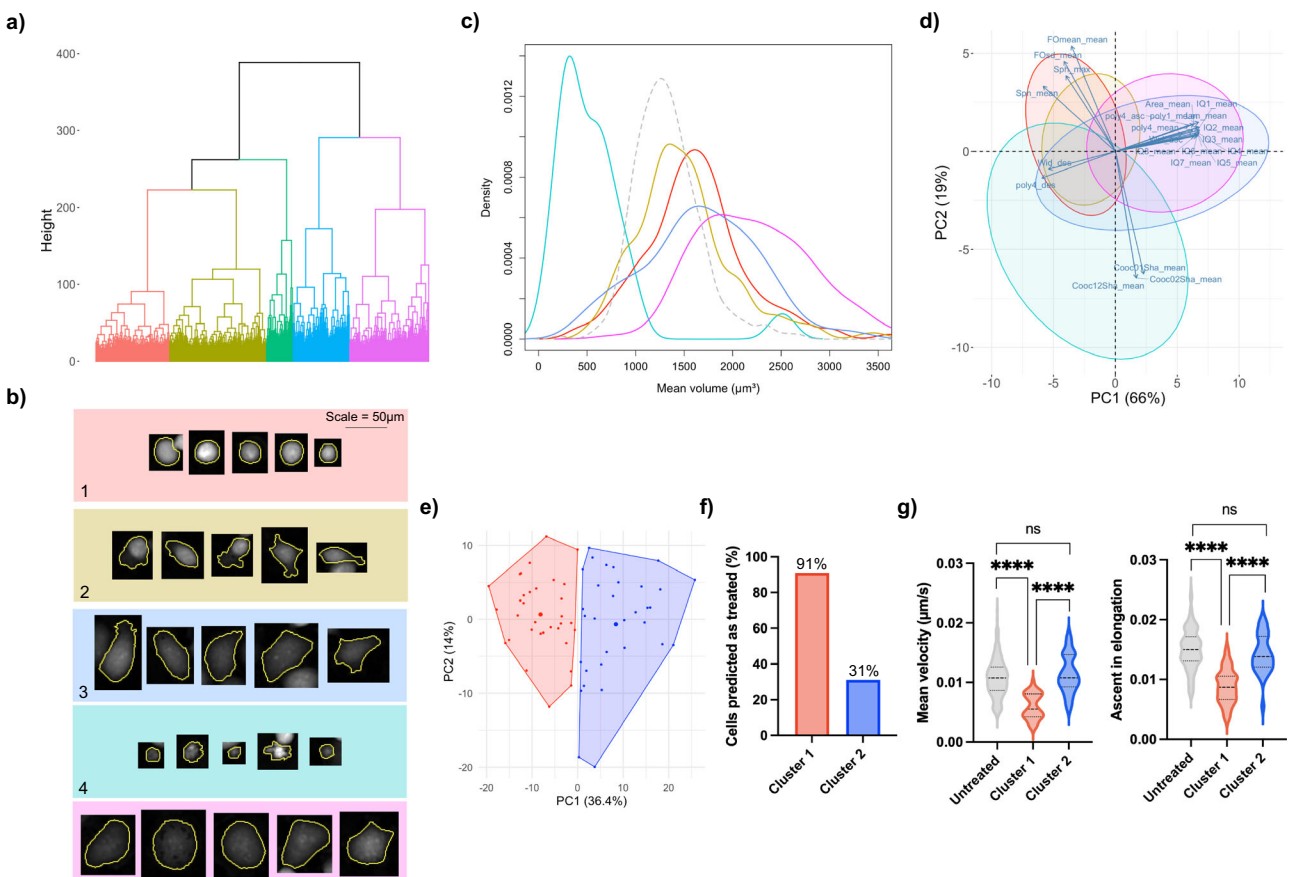

**Fig. 6 | Cluster analysis of treated cells. a** Dendogram obtained from hierarchical clustering of 231Docetaxel treated cells, with 5 clusters coloured. **b** Examples of cells from each cluster with background colours identifying the cluster, note that the scale bar applies to all cell images. Cells within a cluster share similar properties but differ to cells in other clusters. **c** Density plots of mean cell volume, colour-coded according to cluster. The grey, dashed density plot represents 231Docetaxel untreated cells for reference. Cluster 4 (cell debris cluster) has the greatest leftward shift due to cells losing volume upon cell death. Clusters 1 and 2 primarily span the same range of volumes as the untreated set as cells in these clusters have not yet attempted cytokinesis. Clusters 3 and 5 have mean volumes greater than the untreated set as cells in these clusters have continued to grow following failed cytokinesis. **d, e** k-means clustering of 231Doxorubicin test set treated cells. Cells are colour-coded according to which cluster they were assigned. **f** The number of cells predicted as treated for each of the clusters. Cluster 1 was formed of successfully treated cells with 91% (30/33) of cells correctly classified as treated,

whereas cluster 1 formed a subset of non-conforming treated cells, with only 31% (10/32) correctly classified as treated. **g** Increased velocity and ascent in cell elongation are characteristic of untreated cells. These metrics show extremely significant decrease for cells in cluster 1 but no significant difference for cells in cluster 2. Extremely significant differences are observed between cluster 1 and cluster 2, highlighting the presence of subsets within the treated cell population (ns: $p \geq 0.05$, ****: $p < 0.0001$, dashed lines in violin plots are representative of the lower quartile, median and upper quartile). Exact $p$ values were as follows for comparison of mean velocity: Untreated vs. cluster 1: $p = 1.5 \times 10^{-12}$, cluster 1 vs. cluster 2: $p = 1.8 \times 10^{-9}$, untreated vs. cluster 2: $p = 0.3368$. Exact $p$ values were as follows for comparison of ascent in elongation: Untreated vs. cluster 1: $p = 2 \times 10^{-14}$, cluster 1 vs. cluster 2: $5 \times 10^{-8}$, untreated vs. cluster 2: $p = 0.1983$. $p$ values were calculated using a two-tailed, non-parametric Mann–Whitney $U$ test at the 95% confidence interval. Source data for **a, c–e, g** are provided in the Source Data file.

cells following docetaxel treatment due to cell cycle arrest and occasionally cell cycle slippage[23]. Our findings support this with shape and size variables providing the greatest separation for docetaxel treatment in both MDA-MB-231 and MCF-7 cells. Many texture variables were also identified as discriminatory following docetaxel treatment, providing label-free identification of the multiple clusters of high-intensity pixels in treated cells, likely a result of docetaxel-induced multinucleation. We found that at a higher, sub-lethal concentration of 1μM, migration of MDA-MB-231 cells was reduced with variables associated with movement providing greatest discrimination between untreated and doxorubicin treated cells. This is supported by studies that have identified changes in migration of doxorubicin-treated cells, noting that low drug concentrations in fact facilitate increased invasion[24,25].

We found an imbalance in untreated and treated classification accuracy scores, with a greater proportion of treated cells misclassified for all three data sets. This consistent imbalance suggests

the misclassifications are in fact representative of a subset of non-conforming, and potentially chemoresistant, cells. The concept of hidden stratification, where an unlabelled subset performs poorly during classification, has been described previously[26] and poses a challenge in medical research as important subsets (such as rare forms of disease) could be overlooked. Here, the misclassified cells could be of most interest and the ability to identify non-conforming behaviour is precisely what is required from a classifier as treated cells that display behaviour similar to untreated cells could indicate a reduced response to drug treatment. The classification of cells treated with a range of concentrations supported this hypothesis as a greater proportion of cells were classified as untreated at lower drug concentrations, demonstrating that our trained ensemble classifier can be used to quantify drug response, at both single-cell and populational level.

Cluster analysis revealed cell subsets that appear to represent different responses to drug treatment. Heterogeneity of cellular drug

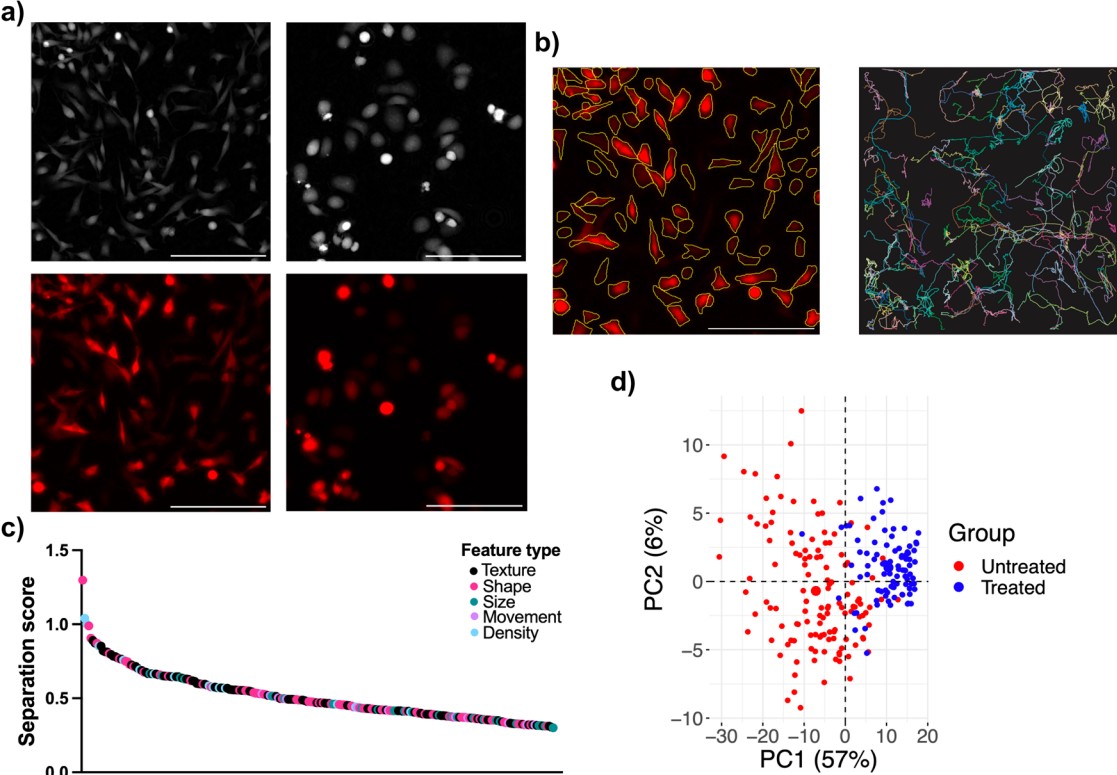

**Fig. 7 | Application of CellPhe to fluorescence images. a** Images taken from cell time-lapses of untreated and 1 μM docetaxel treated MDA-MB-231 cells stably expressing dsRed. Phase and fluorescence images were acquired in parallel. Scale bar = 200 μm. **b** Representative image of Cellpose segmentation on a fluorescent image of MDA-MB-231 cells stably expressing dsRed with cell tracks obtained from TrackMate for untreated MDA-MB-231 cells stably expressing dsRed. Only cell tracks greater than or equal to 50 frames are displayed. **c** Features with separation scores greater than or equal to 0.3, the optimal separation threshold, colour-coded according to feature type. Texture, density, shape and size features provide greatest separation. **d** Principal Component Analysis (PCA) scores plot with points colour-coded according to true class label. Observable separation of classes along PC1 demonstrates that the greatest source of variance within the data arises due to class differences. Only features with separation score greater than or equal to 0.3 were included in PCA. Source data for **c**, **d** are provided in the Source Data file.

response is a commonly reported phenomenon in cancer treatment, yet mechanisms underlying this are not well understood[27]. Analysis of cell volumes showed the mean volume of treated and untreated cells to be comparable for doxorubicin reflecting the fact that this treatment can induce G1, S or G2 cell cycle arrest[28]. However, for docetaxel-treated cells, we found that clusters spanned a range of mean cell volumes beyond those of the untreated set for both cell lines. Clustering allowed identification of three general responses to docetaxel treatment: pre–"cytokinesis attempt", with cells having similar volumes to the untreated MDA-MB-231 population; post-"cytokinesis attempt", where cells were tracked following failed cytokinesis and therefore continued to grow to volumes beyond those of the late stages of the untreated cell cycle; and cell death, with a final cluster, composed primarily of cell debris. Furthermore, giant cell morphology has been linked with docetaxel resistance, a potential cause of relapse in breast cancer patients[9] and through cluster analysis, we were able to identify a potentially resistant subset of very large, treated cells that could be isolated for further investigation.

Our chosen application demonstrated the breadth of quantification and biological insight that can be made by following our workflow, with characterisation of drug response and detection of potentially resistant cells just two of many potential applications for CellPhe. CellPhe offers several benefits for the quantification of cell behaviour from time-lapse images. First, errors in cell segmentation and tracking can be identified and removed, improving the quality of input for downstream data analysis. This is particularly important with machine learning where automation means that such errors can easily be missed, and algorithms consequently trained with poor data. Although different cell lines have different properties that allow segmentation

errors to be recognised, we have shown that ground truth data for a particular cell-line can be re-used for different experiments, in our case, different drug treatments.

Second, cell behaviour is characterised over time by extracting variables from the time series of various features whereas many studies explore temporal changes by collecting data at discrete time points (for example, 0 and 24 hours post-treatment) and using metrics from each static image, missing behavioural changes experienced by cells on a continuous level. With CellPhe, changes over time in features that provide information on morphology, movement and texture are quantified not just by summary statistics but by variables extracted from wavelet transformation of the time series allowing changes on different scales to be identified.

Third, whilst most studies use a limited number of metrics, assessed individually for discrimination between groups[29,30], CellPhe provides an extensive list of metrics and automatically determines the combination that offers greatest discrimination. The bespoke feature selection frequently found the most discriminatory variables to be those with the ability to detect changes in cell behaviour over time. Previous research in this field has focused on identification of cell types from co-cultures[31] for use in automated diagnosis of disease such as cancer. Analysis methods for these studies are often cell-line specific whereas CellPhe's feature selection method is successful in identifying discriminatory variables tailored to different experimental conditions.

Finally, CellPhe uses an ensemble of classifiers to predict cell status with high accuracy and we show that separation scores can be used to identify the variables associated with different cell subsets identified in cluster analysis to explore cell heterogeneity within a population, even when subtle differences are not readily visible by eye.

The interactive, interpretable, high-throughput nature of CellPhe deems it suitable for all cell time-lapse applications, including drug screening or prediction of disease prognosis. We provide a comprehensive manual with a working example and real data to guide users through the workflow step-by-step, where users can interact with each stage of the workflow and customise to suit their own experiments. Here we demonstrated the abundance of information and insight that can be made by following the CellPhe workflow to quantify cell behaviour from QPI images. CellPhe can be used with tracking information from multiple segmentation and tracking algorithms and different imaging modalities, including fluorescence, and would be suitable for all time-lapse studies including clinical applications.

## Methods

### Cell Culture

MDA-MB-231 and MCF-7 cells (American Type Culture Collection [ATCC] catalogue numbers HTB-26 and HTB-22, respectively) were a gift from Prof. Mustafa Djamgoz, Imperial College London. MDA-MB-231 cells and MCF-7 cells were cultured separately in Dulbecco's modified eagle medium supplemented with 5% fetal bovine serum and 4 mM L-glutamine[32]. Fetal bovine serum was filtered using a 0.22μm syringe filter prior to use to reduce artefacts when imaging. Cells were incubated at 37°C in plastic filter-cap T-25 flasks and were split at a 1:6 ratio when passaged. No antibiotics were added to cell culture medium. Cells were confirmed to be mycoplasma-free by 4′,6-diamidino-2-phenylindole (DAPI) method[33]. The molecular identity of MDA-MB-231 and MCF-7 cells was verified by short tandem repeat analysis[34]. Authenticated cell stocks were stored in liquid nitrogen and thawed for use in experiments. Thawed cells were sub-cultured 1–2 times prior to discarding and thawing a new stock to ensure that the molecular identity of cells was retained throughout. In cases where dsRed expressing MDA-MB-231 cells were used, cells were sorted via FACS prior to imaging to enrich for a transfected cell population.

To image the following day, cells were counted and then seeded in a Corning Costar plastic, flat bottom 24-well plate. Cells were seeded at a density of 8000 cells per well with a final volume of 500 μL in each of the 24 wells.

### Pharmacology

Docetaxel (Cayman Chemical Company) was prepared as 5 mg/mL in DMSO and doxorubicin (AdooQ Bioscience) as 25 mg/mL in DMSO; both were then frozen into aliquots. Once thawed, docetaxel and doxorubicin stock solutions were diluted in culture medium to give final working concentrations. Docetaxel dose-response analysis for both MDA-MB-231 and MCF-7 cells involved imaging eight wells treated with the following concentrations of docetaxel: 0 nM, 1 nM, 3 nM, 10 nM, 30 nM, 100 nM, 300 nM, 1 μM, with additional concentrations 3 μM, 10 μM and 30 μM imaged for MDA-MB-231 cells. Doxorubicin dose-response analysis for MDA-MB-231 cells involved imaging eight wells treated with the following concentrations of doxorubicin: 0 nM, 10 nM, 30 nM, 100 nM, 300 nM, 1 μM, 3 μM, 10 μM.

Medium was removed from wells selected to receive treatment 30 mins prior to image acquisition, and 500 μL of desired drug concentration was added to each well. Control wells received a medium change and were treated with DMSO vehicle on the day of imaging to maintain consistent DMSO concentration throughout.

### Image acquisition and exportation

Cells were placed onto the Phasefocus Livecyte 2 (Phasefocus Limited, Sheffield, UK) to incubate for 30 minutes prior to image acquisition to allow for temperature equilibration. One 500 μm × 500 μm field of view per well was imaged to capture as many cells, and therefore data observations, as possible. Selected wells were imaged in parallel for 48 hours at ×20 magnification with 6-minute intervals between frames, resulting in full time-lapses of 481 frames per imaged well. Phase and fluorescence images were acquired in parallel for each well.

For phase images, Phasefocus' Cell Analysis Toolbox® software was utilised for cell segmentation, cell tracking and data exportation. Segmentation thresholds were optimised for a range of image processing techniques such as rolling ball algorithm to remove background noise, image smoothing for cell edge detection and local pixel maxima detection to identify seed points for final consolidation.

The Phasefocus software outputs a feature table for each imaged well. Information on missing frames for tracked cells can be obtained from this table which also provides descriptive features. However, most features are calculated within CellPhe and we only utilise the Phasefocus' features that rely on phase information, these being the volume of the cell and sphericity[35].

For fluorescence images, the TrackMate-Cellpose ImageJ plugin was used for cell segmentation and tracking. Cells were segmented using Cellpose's pre-trained cytoplasm model and image contrast was enhanced prior to segmentation to improve detection of cell boundaries. Once complete, TrackMate feature tables and individual cell ROIs were exported from ImageJ v2.9.0-153t. Prior to use with CellPhe, it was necessary to interpolate TrackMate-Cellpose ROIs to obtain a complete list of cell boundary coordinates. Interpolation of ROIs was performed using a custom ImageJ macro.

### Implementation of CellPhe

Using cell boundary information from Regions of Interest (ROIs) produced by the Phasefocus software or TrackMate, a range of morphological and texture features were extracted for each cell that was tracked for at least 50 frames. Image data were imported into CellPhe using the R package *tiff* v0.1-11. In addition to size and shape descriptors calculated from the cell boundaries, a filling algorithm was used to determine the interior pixels from which texture and spatial features were extracted. The local density was also calculated as the sum of inverse distances from the cell centroid to those of neighbouring cells within three times the cells diameter. A complete list of features together with their definitions is provided in Supplementary table 1.

By considering the position of a cell's centroid on subsequent frames, variables describing the cell's movement were extracted from the images. The current speed of the cell estimated by considering its position in consecutive frames, taking into account any missing frames. The measure provided is proportional to rather than equal to velocity as this would require the rate at which frames were produced to be entered by the user for no gain in discriminatory power. The displacement, or straight line distance between the cell centroid on the current frame and the frame it was first detected in, and the track-length or total path length travelled by the cell up to the current frame, are also calculated. To see how these vary, the quotient current tracklength/current displacement is also calculated.

In addition to volume, calculated using phase information, the size variables determined are cell area, as the number of pixels within (or on) the cell boundary, the length, and width of the cell, determined from the minimal rectangular box that the cell can be enclosed by[36], and the radius, as the average distance of boundary pixels from the cell centroid.

We make use of an imported feature, sphericity, which requires phase information for calculation, but extract a number of other shape features within CellPhe. As well as determining the length and width from the arbitrarily oriented minimum bounding box, we use this to provide a measure of rectangularity as $\max(x,y)/(x+y)$ where $x$ and $y$ are the length and width of the minimal bounding box[37]. We also consider the shape of the cell by calculating the fraction of the minimal box area that the cell area covers and by comparing the number of pixels on the boundary with the total pixels within the cell[37]. Here the

number of boundary pixels is squared in the quotient to avoid the effect of cell size. We also calculate the variance on the distance from the centroid to the boundary pixels, with more circular cells having less variance[37] and an measure of boundary curvature based of the triangle inequality[38]. Finally 4 shape descriptors are obtained from a polygon fitted to the cell boundary, being the mean and variance of both edge length and interior angle[39].

Textural features of each cell are represented in terms of three first order statistics calculated from the pixel intensities within the cell: mean, variance and skewness[40]. For second order texture features, we used grey-level co-occurrence matrices (GLCMs)[41] but, rather than consider the positions of pixels within a cell, we calculated GLCMs between the image of the cell at different resolutions to differentiate textures that are sharp and would be lost at lower resolution from those that are smooth and would remain. This was achieved by performing a two-level 2-D wavelet transform[42] on the pixels within the axis-aligned minimum rectangle containing a cell. GLCMs were then calculated between the original interior pixels and the corresponding values from the first and second levels of the transform as well as between the two sets of transformed pixels (levels 1 and 2). Statistics first described by Haralick[43] were then calculated from each GLCM. We use 14 of the 20 Haralick features described by Löfstedt et al.[44]: Angular Second Moment, Contrast, Correlation, Variance, Homogeneity, Sum Average, Sum Variance, Entropy, Sum Entropy, Difference Variance, Difference Entropy, Information Measure of Correlation 2, Cluster Shade, Cluster Prominence. With three co-occurrence matrices, this gives 42 Haralick features. We calculated spatial distribution descriptors to quantify the uniformity or clustering of cell interior pixels at different intensity levels. $IQn$ is a measure of dispersion calculated for the subset of interior pixels with intensities greater than or equal to the $(n \times 10)$th quantile. Based on a Poisson distribution, for which the mean is equal to the variance, the measure is calculated as the variance divided by the mean, calculated over the pairwise distances between pixels within the $n$th subset. $IQn = 1$ indicates a random distribution whereas a value of $IQn$ less than 1 indicates that the pixels are more uniformly distributed and a value >1 indicates clustering.

Cell tracking provides a time series for each of the 74 features extracted for a cell. The length of the time series depends on how many frames the cell has been tracked for and so differs between cells. In order to apply pattern recognition methods, we extracted a fixed number of characteristic variables for each cell from the time series for each feature. Statistical measures (mean, standard deviation, and skewness) summarise time series of varying length, but may not be representative of changes throughout the time series. Therefore, in addition to summary statistics, we calculated variables inspired by elevation profiles in walking guides, that is, the sum of any increases between consecutive frames (total ascent), the sum of any decreases (total descent) and the maximum value of the time series (maximum altitude gain). Similar variables were calculated for different levels of the wavelet transform of the time series to allow changes at different scales to be considered. The wavelet transform decomposes a time series to give a lower resolution approximation together with different levels of detail that need to be added to the approximation to restore the original time series. Using the Haar wavelet basis[45] with the multiresolution analysis of Mallat[42] allows increases and decreases in the values of the variables to be determined over different time scales. With Haar wavelets, a negative detail coefficient represents an increase from one point to the next, and so we used the sum of the negative detail coefficients to provide the equivalent to total ascent and the sum of the positive detail coefficients as total descent. Rather than an overall maximum, we use the maximum detail coefficient for the transformed time series.

Occasionally the automated cell tracking misses a frame or even several frames, for example when a cell temporarily leaves the field of view. To prevent jumps in the time series, we interpolated values for the missing frames, although these values were not used to calculate statistics. After interpolation, the three elevation variables were calculated from the original time series and three wavelet levels which, together with the summary statistics, provided 15 variables for each feature (Supplementary table 2). The 72 extracted features together with the 2 imported features would have given $74 \times 15 = 1110$ variables in total, but, as one feature, the tracklength or total distance travelled up to the current frame, is monotonically increasing, the total descent is always zero and therefore variables related to tracklength descent were not used. Similarly, as the tracklength and displacement are the same for the first frame and the displacement can never be greater than the tracklength, the maximum value for their quotient will always be 1 and this variable is also not used.

One further variable was introduced to summarise cell movement as the area of the minimal bounding box around a cell's full trajectory. This area will be large for migratory cells and small for cells whose movement remains local for the duration of the time series. If, within a cell's trajectory, $\min X$ and $\min Y$ are the minimal $X$ and $Y$ positions respectively with $\max X$ and $\max Y$ the corresponding maximal positions, then the trajectory area is defined as

$$\text{trajectory area} = (\max X - \min X) \times (\max Y - \min Y). \qquad (1)$$

Thus, a total of 1106 characteristic variables were available for analysis and classification.

To improve characterisation of cellular phenotype, we only included cells that were tracked for at least 50 frames in our analyses. Whilst the majority of these cells were correctly tracked, others had segmentation errors, with confusion between neighbouring cells, missing parts of a cell or multiple cells included.

In order to increase the reliability of our results, we developed a classification process to identify and remove such cells prior to further analysis. Cells (both treated and untreated) were classified by eye to provide a training data set. Due to class imbalance, with the number of segmentation errors far less than the number of correct segmentations, the Synthetic Minority Oversampling Technique (SMOTE)[46] was performed using the *smotefamily* package v1.3.1 in R, with the number of neighbours $K$ set to 3, to double the number of instances representing segmentation errors.

The resulting data set with all 1111 variables was used to train a set of 50 decision trees using the *tree* package v1.0-4.2 in R with default parameters. For each tree, the observations from cells with segmentation errors were used together with the same number of observations randomly selected from the correctly segmented cells to further address class imbalance. For each cell, a voting procedure was used to provide a classification from the predictions of the 50 decision trees. To minimise the number of correctly tracked cells being falsely classified as segmentation errors, this class was only assigned when it received at least 70% of the votes (i.e., 35). To add further stringency, the training of 50 decision trees was repeated ten times and a cell only given a final classification of segmentation error if predicted this label in at least five of the ten runs. MDA-MB-231 cells that were not used for training formed an independent test set. All cells either manually labelled as segmentation error or predicted as such were excluded from further analyses.

After removing segmentation errors, the remaining data were used to form training and test sets for the classification of untreated and treated cells. Training sets were balanced prior to classifier training to mitigate bias and data from cells in the independent test sets were never used during training.

A separate classifier was trained for each cell line−treatment combination, as shown in Table 3 and feature selection performed to

**Table 3 | The three data sets used in this study with the number of cells in training and test sets used for untreated vs treated classification**

| Data set | Cell line | Treatment | Training set | Test set |
|---|---|---|---|---|
| 231Docetaxel | MDA-MB-231 | 30 µM Docetaxel | Untreated: 646 | Untreated: 913 |
| | | | Treated: 600 | Treated: 300 |
| 231Doxorubicin | MDA-MB-231 | 1 µM Doxorubicin | Untreated: 213 | Untreated: 191 |
| | | | Treated: 215 | Treated: 60 |
| MCF7Docetaxel | MCF-7 | 1 µM Docetaxel | Untreated: 200 | Untreated: 441 |
| | | | Treated: 200 | Treated: 128 |

determine the most appropriate variables in each case. Each variable was assessed using the group separation, $S = V_B/V_W$, where $V_B$ is the between-group variance:

$$V_B = \frac{n_1(\bar{x}_1 - \bar{\bar{x}})^2 + n_2(\bar{x}_2 - \bar{\bar{x}})^2}{(n_1 + n_2 - 2)} \tag{2}$$

and $V_W$ is the within-group variance:

$$V_W = \frac{(n_1 - 1)s_1^2 + (n_2 - 1)s_2^2}{(n_1 + n_2 - 2)}. \tag{3}$$

Here $n_1$ and $n_2$ denote the sample size of group 1 and group 2 respectively, $\bar{x}_1$ and $\bar{x}_2$ are the sample means, $\bar{\bar{x}}$ the overall mean, and $s_1^2$ and $s_2^2$ are the sample variances. The most discriminatory variables were chosen for a particular data set by assessing the classification error on the training data to optimise the threshold on separation. Starting with a threshold of zero, the nth separation threshold was minimised such that the classification error rate did not increase by more than 2% from that obtained for the (n−1)th threshold. The aim here was to reduce the risk of overfitting by only retaining variables achieving greater than or equal to this threshold for the next stage of classifier training.

Data were scaled to prevent large variables dominating the analysis and ensemble classification used to take advantage of different classifier properties. The predictions from three classification algorithms, Linear Discriminant Analysis (LDA), Random Forest (RF) and Support Vector Machine (SVM) with radial basis kernel were combined using the majority vote. Model performance was evaluated by classification accuracy, taking into account the number of false positives and false negatives. All classification was performed in RStudio V1.2.5042[47] using open-source packages. LDA was performed using the *lda* function from the *MASS* library[48], SVM classification used the *svm* function from the package *e1071* v1.7-12[49] with a radial basis kernel and the package *randomForest* v4.7-1.1[50] was used to train random forest classifiers with 200 trees and 5 features randomly sampled as candidates at each split.

Both hierarchical clustering and *k*-means clustering were used to investigate subgroups within single-class data sets (i.e. treated and untreated cells separately). Data were scaled prior to clustering and analyses performed in R. Hierarchical clustering was implemented with the *factoextra* package v1.0.7[51] using the *hcut* function to cut the dendrogram into *k* clusters. Agglomerative nesting (AGNES) was used with Ward's minimum variance as the agglomeration method and the Euclidean distance metric to quantify similarity between cells. *k*-means clustering was performed using the R *stats* package v4.1.3, with the number of random initial configurations set to 50. The number of clusters *k* was chosen to obtain clusters with meaningful interpretation. Similarities and differences between clusters were identified through evaluation of separation scores to determine discriminatory features, as well as through observation of cells within each cluster by eye.

## Statistics and reproducibility

All tests of statistical significance within this study were performed using Graphpad Prism 9.1.0 (GraphPad Software, San Diego, CA). Data were tested for normality using the D'Agostino & Pearson test. Parametric tests (*t* tests and *F* tests) were used where suitable with non-parametric Mann-Whitney *U* tests in place of *t* tests where data did not follow a normal distribution. Results were considered significant if $p < 0.05$. Levels of significance used: $* < 0.05$, $** < 0.01$, $*** < 0.001$, $**** < 0.0001$. Full details of statistical tests used for each analysis are provided in the figure legend for the corresponding figure.

Three data sets were used to demonstrate our pipeline for the classification of untreated and treated cells. For brevity we use abbreviations throughout to refer to each data set, for example, 231Docetaxel is a data set consisting of MDA-MB-231 cells, both untreated and treated with 30 µM docetaxel. This is the main data set used to develop the methods, with a training data set compiled from 6 experiments performed on different days and an independent test data set compiled from a further 3 experiments, also performed on separate days and by a different individual.

We validate our methods using two further data sets, the 231Doxorubicin and MCF7Docetaxel data sets, details of which are given in Table 3. This table also includes details of the number of cells within each training and test set. We show that the classification pipeline can be successfully reproduced using fewer experimental repeats for the 231Doxorubicin and MCF7Docetaxel data sets. The 231Doxorubicin training set consists of data from one experiment with a further, independent experiment performed on a separate day used as a test set. Training and test sets for MCF7Docetaxel are from the same two experiments, with random sampling used to produce independent training and test sets. Each training data set contains a balanced number of untreated and treated cells, treated with a single drug concentration. We selected 30 µM docetaxel and 1 µM doxorubicin for the experiments with MDA-MB-231 cells as the optimal doses with which to induce changes in cell morphology and migration without inducing cell death. However, a lower concentration (1 µM) of docetaxel was used for MCF-7 cells as we found that this induced similar morphological and dynamical changes to those induced by higher concentrations but with reduced cell death (Table 3).

## Reporting summary

Further information on research design is available in the Nature Portfolio Reporting Summary linked to this article.

# Data availability

All data used to produce the results in the manuscript, including separate data that will allow the user to follow the worked example in the CellPhe user guide, are available from the Dryad Database[52] https://doi.org/10.5061/dryad.4xgxd25f0. This includes the file `example_data.zip` which contains all the data required to follow the worked example. A video `CellPhe_GUI_demo_vid.mov` that explains

how to use the GUI is available from https://zenodo.org/record/7674584#.ZAJYBOzP0o8. Source data are provided with this paper.

## Code availability

The source code for algorithms developed during this research has been deposited in GitHub, linked from https://zenodo.org/record/7620171#.ZAJZMuzP0o8[53]. The interactive CellPhe GUI can be accessed at https://cellphegui.shinyapps.io/app_to_host/.

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

## Acknowledgements

We would like to thank Dr. Jon Pitchford for his ongoing valuable advice and Dr. Fiona Frame for providing initial data sets. We would also like to thank the University of York Bioscience Technology Facility—Imaging and Cytometry Team for the helpful technical assistance they provided throughout the project. We express gratitude to Phasefocus UK for the Livecyte and CATbox systems that were used to acquire and export all time-lapse data presented here, and for their technical support throughout. L.W. is supported by a BBSRC NPIF Ph.D. studentship, grant number: BB/S507416/1.

## Author contributions

Conceptualisation: W.B., P.O.'T., J.W., and L.W.; cell culture, pharmacology, and imaging: L.W. and A.L.; data analysis and validation: L.W. and J.W.; software development J.W., L.W, S.L., and K.M; supervision: J.W., W.B., and P.O'T.; writing-original draft preparation, L.W., and J.W.; writing-review and editing, W.B. and P.O.'T.

## Competing interests

The authors declare no competing interests.
