## [Peer Review File · Nature Communications]

The CellPhe toolkit for cell phenotyping using time-lapse imaging and pattern recognitionThis manuscript has been previously reviewed at another journal that is not operating a transparent peer review scheme. This document only contains reviewer comments and rebuttal letters for versions considered at *Nature Communications*.

REVIEWER COMMENTS

Reviewer #1 (Remarks to the Author):

Unfortunately, while in general their approach seems much improved, it's still not at a level I personally feel merits publication - in attempting to follow the manual, I could not get past page 2, because they do not provide either a) example files on how to perform the optional segmentation correction step (they do somewhat explain how to make them, but not in a way where I believe a beginner could successfully do so) or b) example files OR descriptions of what the files should look like to perform the actual classification steps (these files listed in the manual are not provided in the GitHub). The paper and the manual both claim there is a GUI, but neither actually gives instructions for accessing it and I could not figure out how to do so (the provided video shows what to do in the software once you have the GUI running, but not how to actually initialize the GUI in the first place) - I'm reasonably certain the Shiny app actually hasn't been committed to the repository at all. If this is meant to be an approach that others can use, it unfortunately still needs a lot more documentation, example data, and user friendliness before it rises to the level that merits being published in Nature Communications.

Reviewer #2 (Remarks to the Author):

The paper "CellPhe: a toolkit for cell phenotyping using time-lapse imaging and pattern recognition" of Wilson et al. describes a pipeline and a tool for the characterization of cells including the search for group differences. The pipeline assumes an already existing segmentation and tracking step that will massively influence the quality of all following pipeline modules. The main merits of the paper are (1) the potentially very systematic and tool-supported methodology to find interpretable group differences via deletion of erroneous cell segmentation and tracking, feature generation, feature selection, classification and search for subgroups and (2) the interesting use cases for the characterization of chemotherapeutics for an in-vitro setting using selected cell lines.

The main advance of the revised version is the integration of all parts in R. In addition, the paper has been improved in many smaller points suggested by the reviewers in the first revision round. The main recommended point of a possible tighter coupling with segmentation and tracking algorithms was not addressed. From my point of view, this is where the greatest long-term potential of the method lies, although this would require close embedding in the tools mentioned. However, the authors deliberately decided against it.

From my point of view, it is only a borderline paper for Nature Communications, because although many useful elements are combined for a use case, the essential novelty value is limited. My recommendation is still a transfer towards Scientific Reports or a similar journal. In the end though, that's just a recommendation and I would accept a majority opinion from the other reviewers and the editorial board.

Typo in the Suppl. Material
Texure -> Texture (2x)

Reviewer #3 (Remarks to the Author):

I am happy with the changes made to the manuscript.
All my points were addressed in the author's response.

The additional example demonstrating that the system can work on fluorescence images as well enhances the utility of the software.

It looks like the new R package is significantly easier to use than the previous version.

RESPONSE TO REVIEWERS' COMMENTS

Reviewer #1 (Remarks to the Author):

Unfortunately, while in general their approach seems much improved, it's still not at a level I personally feel merits publication - in attempting to follow the manual, I could not get past page 2, because they do not provide either a) example files on how to perform the optional segmentation correction step (they do somewhat explain how to make them, but not in a way where I believe a beginner could successfully do so) or b) example files OR descriptions of what the files should look like to perform the actual classification steps (these files listed in the manual are not provided in the GitHub). The paper and the manual both claim there is a GUI, but neither actually gives instructions for accessing it and I could not figure out how to do so (the provided video shows what to do in the software once you have the GUI running, but not how to actually initialize the GUI in the first place) - I'm reasonably certain the Shiny app actually hasn't been committed to the repository at all.

We profusely apologise for the problems associated with our code and data availability statements. We have now ensured that these are correct and that all data files, the CellPhe GUI and demonstration video are readily available. All code as well as the CellPhe user guide are available on Github (<https://github.com/uoy-research/CellPhe>) and the GUI is hosted at https://cellphegui.shinyapps.io/app_to_host/. All data used to produce the results in the manuscript, including separate data that will allow the user to follow the worked example in the CellPhe user guide, are available from <https://doi.org/10.15124/936b6b09-a341-40ee-b08f-c049316ac247>. Here, the file example_data.zip contains all the data required to follow the worked example and the file CellPhe GUI demo vid.mov is a video that explains how to use the GUI. Furthermore, we have added additional examples of ground truth segmentation data sets and documentation on how to create these to further aid a user who wishes to make use of the optional segmentation error classification step within their own work.

If this is meant to be an approach that others can use, it unfortunately still needs a lot more documentation, example data, and user friendliness before it rises to the level that merits being published in Nature Communications.

Following Reviewer 1's comments we have made use of a non-expert microscopist to trial the CellPhe R package and GUI using the CellPhe manuscript and user guide with no further external instructions. This user was able to successfully complete the worked example within the CellPhe user guide and GUI demonstration video despite no familiarity with coding in R and provided useful feedback on additional documentation that would further assist users with limited coding experience (for example, additional documentation on importation of TrackMate data) which we have now added to the user guide.

In summary, following the Reviewer's helpful feedback, we have enhanced the documentation and example data to facilitate accessibility of CellPhe to users at all levels. The improved accessibility will enable this tool will have a significant impact in the field of microscopy image analysis and thus merits publication in Nature Communications.

Reviewer #2 (Remarks to the Author):

The paper “CellPhe: a toolkit for cell phenotyping using time-lapse imaging and pattern recognition” of Wilson et al. describes a pipeline and a tool for the characterization of cells including the search for group differences. The pipeline assumes an already existing segmentation and tracking step that will massively influence the quality of all following pipeline modules. The main merits of the paper are (1) the potentially very systematic and tool-supported methodology to find interpretable group differences via deletion of erroneous cell segmentation and tracking, feature generation, feature selection, classification and search for subgroups and (2) the interesting use cases for the characterization of chemotherapeutics for an in-vitro setting using selected cell lines.

We thank the Reviewer for highlighting the important merits of our paper, including systematic and tool-supported methodology and widely applicable use cases.

The main advance of the revised version is the integration of all parts in R. In addition, the paper has been improved in many smaller points suggested by the reviewers in the first revision round.

We thank the Reviewers for their invaluable feedback on our initial version of CellPhe. We agree that the integration of all parts in R and additional enhancements including the GUI and supporting documentation have significantly improved our manuscript.

The main recommended point of a possible tighter coupling with segmentation and tracking algorithms was not addressed. From my point of view, this is where the greatest long-term potential of the method lies, although this would require close embedding in the tools mentioned. However, the authors deliberately decided against it.

We agree that integration of CellPhe with pre-existing segmentation and tracking software is an important advancement and CellPhe does allow importation of data from leading tracking software, TrackMate, which itself imports output from various popular segmentation software such as Ilastik and CellPose. We would be happy to collaborate with authors of segmentation and tracking algorithms or indeed to work with users to ensure that the particular format of their tracking data is accommodated. Following the Reviewer’s comments we have made contact with Jean-Yves, creator of TrackMate, to discuss the possibilities of collaboration. However, embedding into additional image analysis tools is beyond the scope of the current manuscript. Nonetheless, we argue below that CellPhe provides a novel and important contribution, worthy of publication in Nature Communications.

From my point of view, it is only a borderline paper for Nature Communications, because although many useful elements are combined for a use case, the essential novelty value is limited. My recommendation is still a transfer towards Scientific Reports or a similar journal. In the end though, that's just a recommendation and I would accept a majority opinion from the other reviewers and the editorial board.

We strongly believe that the work presented here is novel and will be of immense value to the live-cell imaging community. Although some cell tracking software does provide quantification of basic features, the time series data are only available for viewing or output. **No other software to date characterises these time series in terms of variables that can be used with machine learning algorithms for unsupervised or supervised analysis of cell phenotypes.** We demonstrate here that an abundance of phenotypic data can be hidden within microscopy images but that the complexity of extraction and analysis of this information can result in overlooking this important information. CellPhe is the first tool to facilitate the extraction and analysis of thousands of time series metrics for cell type characterisation, classification and identification of heterogeneous populations. We therefore argue that this contribution is worthy of publication in a journal with wide reach, such as Nature Communications.

Typo in the Suppl. Material
Texure -> Texture (2x)

We thank the Reviewer for highlighting these typos, which have now been fixed.

Reviewer #3 (Remarks to the Author):

I am happy with the changes made to the manuscript.
All my points were addressed in the author's response.
The additional example demonstrating that the system can work on fluorescence images as well enhances the utility of the software.
It looks like the new R package is significantly easier to use than the previous version.

We thank the Reviewer for their time and for their useful initial comments that helped us to reshape and improve CellPhe. We are pleased to note that the Reviewer sees the value of the extension to fluorescence images in our revised manuscript. We believe this enhancement will significantly increase the impact of our tool. We agree that the CellPhe R package is significantly easier to use, with both the R package and the GUI now being accessible to users of all levels and disciplines.

REVIEWERS' COMMENTS

Reviewer #1 (Remarks to the Author):

I thank the authors for updating the documentation and adding the link to the shiny app. I am very glad to hear they had a naive user run the protocol and updated it in response - such exercises are critical to making tools useful to the broader community.

I am now satisfied that the tool and documentation meet the level of publication, and thank the authors for their extensive work from the original version of the tool to now!

RESPONSE TO REVIEWERS' COMMENTS

Reviewer #1 (Remarks to the Author):

I thank the authors for updating the documentation and adding the link to the shiny app. I am very glad to hear they had a naive user run the protocol and updated it in response - such exercises are critical to making tools useful to the broader community.

I am now satisfied that the tool and documentation meet the level of publication, and thank the authors for their extensive work from the original version of the tool to now!

RESPONSE

We thank the reviewer for their time and are very pleased that they consider the software suitable for publication.